One step further in biomechanical models in palaeontology: a nonlinear finite element analysis review

http://orcid.org/0000-0001-9852-7027 Marcé-Nogué Jordi 1 2 jordi.marce.nogue@uni-hamburg.de
1 Department of Mechanical Engineering, Universitat Rovira i Virgili Tarragona , Tarragona, Catalonia , Spain
2 Institut Català de Paleontologia, Universitat Autònoma de Barcelona , Cerdanyola del Vallès, Catalonia , Spain
Hutchinson John
Electronic publication date: 2022 Aug 8
Publication date: 2022
Volume: 10
Electronic Location ID: e13890
Received 2022 May 13; Accepted 2022 Jul 21
Copyright: © 2022 Marcé-Nogué
Copyright year: 2022
Copyright holder: Marcé-Nogué
License: This is an open access article distributed under the terms of the Creative Commons Attribution License, which permits unrestricted use, distribution, reproduction and adaptation in any medium and for any purpose provided that it is properly attributed. For attribution, the original author(s), title, publication source (PeerJ) and either DOI or URL of the article must be cited.
License URL: https://creativecommons.org/licenses/by/4.0/

Keywords: Finite element analysis, Palaeontology, Anthropology, Non-linear methods, Functional morphology

Funding: CERCA programme (ICP) from the Generalitat de Catalunya PID2020-117118GB-I00, MCIN/AEI/10.13039/501100011033 This work was supported by the Serra-Hunter (URV) and the CERCA programme (ICP) from the Generalitat de Catalunya and the research project PID2020-117118GB-I00 funded by MCIN/AEI/10.13039/501100011033. The funders had no role in study design, data collection and analysis, decision to publish, or preparation of the manuscript.

==============================
Finite element analysis (FEA) is no longer a new technique in the fields of palaeontology, anthropology, and evolutionary biology. It is nowadays a well-established technique within the virtual functional-morphology toolkit. However, almost all the works published in these fields have only applied the most basic FEA tools i.e., linear materials in static structural problems. Linear and static approximations are commonly used because they are computationally less expensive, and the error associated with these assumptions can be accepted. Nonetheless, nonlinearities are natural to be used in biomechanical models especially when modelling soft tissues, establish contacts between separated bones or the inclusion of buckling results. The aim of this review is to, firstly, highlight the usefulness of non-linearities and secondly, showcase these FEA tool to researchers that work in functional morphology and biomechanics, as non-linearities can improve their FEA models by widening the possible applications and topics that currently are not used in palaeontology and anthropology.

Introduction

Computational biomechanics represents the application of computational tools in mechanical problems to study biological systems. During the last decade, computational methods such as finite element analysis (FEA) have been widely used in the field of palaeontology to study biomechanical behaviour of a vast array of fossil species (Rayfield, 2007). However, almost all works published in the field have applied the most basic FEA capabilities, i.e. liner materials in static structural problems, where we can easily define the relationship between physical parameters by means of linear equations. This kind of equations are easy to solve using direct solvers as they exhibit a low computational cost that is a direct response of the size of the finite element mesh. A larger number of nodes in a finite element mesh involves more unknowns in the equation and, consequently, more time is required to solve the mathematical system of equations. Nevertheless, mathematical nonlinearities are natural in physical laws and the assumption of linearity and staticity is a simplification of reality to make the problem easy to solve. This is because a nonlinear system is characterized with an output that is not proportional to the change of the input. The inclusion of this complexity entails an increase in the computational cost of solving the equations and the need of using iterative solvers. Consequently, palaeontologists have primarily used linear approximations and static problems because these are easier to calculate, computationally faster and solutions can be superposed on each other, hence avoiding an iterative process. In addition, most palaeontologists do not have a deep background on how FEA problems are defined and solved, and many are not aware of the potential of non-linear modelling. For example, including non-linearities in paleoanthropological models can improve the results obtained when modelling skulls by considering sutures, thus making the behaviour of the model closer to reality (Tanaka et al., 2000). They can also be useful in understanding the failure of slender and thin bones or when trying to understand the taphonomic deformations that affected some fossil material by carrying out retrodeformation procedures. Nowadays, most of both commercial and non-commercial FEA packages can solve non-linearities and some examples are already published in living species or in biomedical studies involving humans. Therefore, it makes sense to explore the possibilities that non-linear FEA could provide to the palaeontological and anthropological communities, as these approaches would allow them to explore a broader range of scientific questions, including topics that are currently unsolved or not modelled with enough accuracy. Consequently, the aim of this review is to, firstly, showcase when non-linearities can be useful in functional morphology and secondly, to introduce these methods to researchers that work biomechanics as a way to improving their FEA models by showing them examples that currently are not used in this field. Therefore, this review can be of interest for palaeontologists that seek new ideas in their research, functional morphologists that want to be one step beyond in their research and other researchers who work in the life sciences or in computational mechanics that want to know the state-of-the-art in non-linear FEA applied to biomechanical models.

Search methodology

The literature cited in this text is based on a personal selection made by the author to reliably characterise the methods described in the text. A previous search in the Google Scholar database was done to select the appropriate references for the examples. Different keywords were used in each analysed case to fit the expected search. The final selection of the references was based on covering- if possible- diverse animal families, different morphologies, or different fields.

Discovering all the fea elements: solids, shells, plates, beams, springs, and trusses

Finite element analysis (FEA) is the mathematical way to solve problems of elasticity in complex geometries by dividing the geometry in small elements where the equations are easy to solve (Zienkiewicz & Taylor, 1981). The equations of elasticity relate the external forces applied in a body to understand how it deforms and how the inner forces are distributed inside them (Timoshenko & Gere, 1961). The underlying premise of the method is that a complex geometry can be subdivided into a mesh consisting of a finite number of elements in which the respective equations are approximately solved (Marcé-Nogué et al., 2015). This method has been widely used in palaeontology, anthropology and functional morphology mostly because we can easily digitize bony structures (Lautenschlager, 2016) to then apply FEA to the obtained geometries. It cannot be omitted that the generated models are not literal representations of reality, but they still may be useful for answering scientific questions (Anderson et al., 2012). Following the idea of simplification there are different kinds of elements that we can use when we are creating a FEA model (Fig. 1). The use of some of these elements will result in a greater degree of simplification from reality than others because different simplifications are assumed in terms of geometry and mechanical behaviour, but also because it involves a reduction of the complexity of the mathematical equations and the computational time.

Figure 1 Examples.

Examples of bar elements (Marcé-Nogué & Liu, 2020) shell elements (Püschel et al., 2020a), plane elements (Marcé-Nogué et al., 2020) and solid finite elements (Zhou et al., 2017).

The elements beam and spring or truss are used when the original geometry is a line or can be assumed as a line and the model is defined either in the 3D space or in 2D. The primary difference between these elements is that beam elements follow beam theory (Timoshenko, 1955), which enables the calculation of the loads and deflection of beams subjected to outer forces (including bending, shear, torsion and axial forces). Springs and truss elements, in contrast, are only designed to handle tensile and compressive forces in the axial direction of the element. Examples can be found when simplifying the skull of reptiles and mammals to a beam model (Preuschoft & Witzel, 2002). However, they are not widely used because these models can be solved most of the time by hand without the need of a computer. However, the use of springs or truss are widely extended as a complement of the model when it is necessary to include tendons, ligaments, or other complementary biological structures of the main model. For example, FEA models of the carpal bones include spring elements to model the presence of ligaments between bones (Gíslason et al., 2017).

Shell and plate elements are commonly used when the geometry can be assumed as a surface with constant thickness and the model is defined in 3D space. The difference between shells and plates are that shells are used in curved surfaces and plates only in plane surfaces. Mechanically speaking, both shells and plates can handle bending, but shells develop membrane forces whereas plates do not. This means that shell elements include the membrane effects of resistance to compressive and tensile forces, whereas plates do not. In most of the biological models modelling bone structures, shells have been the preferred option in front of plates. An example of shell elements can be found in works analysing carpal bones (Püschel et al., 2020a) or talar morphologies (Püschel et al., 2020b) because they have a tiny layer of cortical bone with cancellous bone inside where the cortical bone can be assumed as a surface or when modelling something thin as dragonfly wings (Rajabi et al., 2016a).

Another assumption that may further reduce the dimensions of the problem may be simplifying to a surface that lies in a 2D plane using plane elements. I suggest calling them as plane elements because these elements are not really in 2D since they have a constant thickness and use the equations of plane elasticity. When solving the equations of elasticity, plane elasticity refers to the study of specific solutions of the elastic problem in bodies that are surfaces with a constant thickness that are lying in a plane and the forces you apply should lie in this plane. Examples of plane models can be widely found in studies focused on mammal mandibles (Lautenschlager et al., 2020; Marcé-Nogué et al., 2020) or in dinosaurs and other fossils (Neenan et al., 2014; Ma et al., 2021). Plane models also can be useful when modelling other morphologies such as trilobites (Esteve et al., 2021), claws (Patiño, Pérez Zerpa & Fariña, 2019), beaks (Miller et al., 2020) or teeth (Ballell & Ferrón, 2021).

It is important to point out the differences between shell, plate, and plane elements. First, shell elements are not lying in a plane whereas plane and plate elements are. Secondly, plate element allows forces that are not in the plane, like perpendicular forces, supporting bending whereas plane elements do not. This difference can be seen in previous FEA modelling studies of several temnospondyl amphibians (Fortuny et al., 2012) or crocodylomorphs (Pierce, Angielczyk & Rayfield, 2009) where the forces applied are perpendicular to the flat surface of the skull during bilateral cases where plate elements where used.

Finally, solid elements are used when the geometry is a volume, and the model is built in the 3D space. They have been the most widely used in palaeontology and anthropology because they can be easily created from after digitizing a real geometry using CT scanning, photogrammetry, laser scanners, among others. Examples can be found in FEA models of mandibles which have been modelled in 3D (e.g. (Zhou et al., 2019)), unlike the simpler plane models described above. Solid elements can also be found in models of skulls (Zhou et al., 2017), teeth (Benazzi et al., 2012) and a broad range of postcranial (Püschel & Sellers, 2016; McCabe et al., 2017; Bucchi et al., 2020) and other biological structures (Nagel-Myers et al., 2019; Bicknell et al., 2021; Klunk et al., 2021; Krings, Marcé-Nogué & Gorb, 2021).

Non-linearities in fea models

In general, a nonlinear system is a mathematical system in which the change of the output variable is not proportional to the change of the input variable and, consequently, the equations cannot be written as a linear combination of the unknown variables (Kim, 2015). Therefore, the equations of nonlinear systems are more difficult to solve. A common strategy to deal with them is to approximate the system by linear equations performing multiple iterations to converge to the correct solution (Fig. 2). On the contrary, problems are linear when the output variable is proportional to the change of the input variable. Linearities are found in elastic materials (i.e., following the Hooke’s Law) or when using the small strain theory. This theory is applied when deformations are much smaller than the body dimensions. Therefore, the deformed and undeformed configurations of the body under analysis are assumed to be the same. The equations of continuum mechanics are considerably simplified when applying this assumption by ‘linearising’ (i.e., making linear) the problem to be solved. Non-linearities can be originated by different phenomena in these systems:

Figure 2 Convergence.

Relationship between external force (F) applied in a body and displacement (u) in (A) linear problem (B) non-linear problem. K is the stiffness of linear models. Ki is the predicted stiffness in non-linear models to reach the convergence.

Material non-linearity: When a material is non-linear, the strain it experiences is not proportional to the stress applied i.e., the material does not conform to Hooke’s Law. This situation occurs in plastic or hyperelastic materials where the relationship between stress and strain does not follow a lineal proportion.

Large deformation non-linearity: The so-called finite strain theory, large strain theory, or large deformation theory is used when strains are large enough to invalidate the assumptions of the small strain theory, which is the theory commonly used in linear elastic problems. In this case, the deformed and undeformed configurations of the body under analysis are notably different, requiring a clear distinction between them in the formulation that, consequently, also affects the relation between stress and strain in the constitutive equation. This theory is common in elastomers and soft tissues and needs to be used when modelling hyperelastic materials.

Large displacement non-linearity: Also called as geometrical non-linearity, assumes small strains but large rotations and displacements. In the geometrically linear case, the forces are applied in the undeformed geometry when solving the model whereas in the geometrically nonlinear cases, the applied forces depend on the deformed upcoming geometry. It involves an iterative solution accounting for the displacements and needs to be considered when analysing buckling.

Non-linear contacts: Separate surfaces of two bodies are in contact without overlapping in such a way that they become mutually tangential. Depending on the relationship between these two surfaces, contacts that allow the separation in the perpendicular direction require a nonlinear solution because there are unknowns at the start of the solving process i.e., where and which force is applied.

The mathematical methods applied to solve general nonlinear functions are all iterative starting from an initial estimation. The solution is obtained by solving iteratively a linearization of the non-linear system in different steps towards the convergence of the solution. Different methods are available depending on the procedure of calculating the increment of the steps: the Newton-Raphson method, the incremental secant method or the incremental force method among others (Kim, 2015). Therefore, the computational cost of the solving procedure of a nonlinear FEA model is now not only affected by the size of the mesh, but also affected by the number of iterative resolutions before convergence.

Non-linear materials: hyperelasticity and plasticity

Non-linear materials are materials in which the constitutive equation that defines their behaviours establishes a relationship between stress and strain that is not proportional to a constant. Typical material non-linearities can be found in phenomena such as plasticity and hyperelasticity. Plasticity describes the deformation of a material undergoing non-reversible changes of shape in response to applied forces. In a typical stress-strain curve for a plastic material there is a linear elastic region which satisfies Hooke’s law and a plastic region before fracture that can also follow a linear law or can be defined using different linear sections (Fig. 3). The transition from elastic behaviour to plastic behaviour is called yield and a non-linear solution is required because the solver needs to discover if the body is in the plastic region or not. The total strain is defined by εtotal=εelastic+εplastic and the value of stress will depend on the value of this total strain. In a biological context, plasticity can be found in trabecular bone formulations to capture tension-compression asymmetry in the yield strength (Gupta et al., 2007) or, more generally, in studies where a permanent deformation or plasticity is assumed in cortical bone or other biological materials such as dentin, enamel or nacre (An, 2016). In materials such as dentine or enamel, plasticity represents the extremes of the loading environment rather than everyday behaviour.

Figure 3 Materials.

Constitutive equations between stress (σ) and strain (ε) for (A) plastic materials using a bilinear model and (B) hyperelastic materials. In a plastic material εT is the total strain when there is elastic strain (εE) and εt plastic strain; E is the Young Modulus. In a hyperelastic material W is the strain energy and Eij the components of the strain tensor.

A hyperelastic material is one that shows extreme elastic behaviour, in that it can return to its original shape even after experiencing very high strains. They are ideal elastic materials in which the stress-strain relationship is non-linear because derives from a strain energy function instead of Hooke’s law (Fig. 3). Moreover, these materials use the large deformation theory already mentioned above. However, the response of the material is not plastic because deformations are fully recoverable. Typical formulations of hyperelastic materials are, among others, phenomenological descriptions of observed behaviour in Mooney–Rivlin and Ogden formulations or equations describing the underlying structure of the material in the Neo–Hookean model (Ogden, 1984). Hyperelastic formulations are common in soft tissues such as ligaments or tendons (Shearer, 2015). Specifically, they can be found in the periodontal ligament (PDL) (Bucchi et al., 2019), muscles such as the pelvic floor (Stansfield et al., 2021), the abdominal muscle (Tuset et al., 2019) or generic muscular tissues (Hedenstierna, Halldin & Brolin, 2008), skin (Ito et al., 2022), corneas (Shan et al., 2010), cartilage (Pataky, Koseki & Cox, 2016), the temporomandibular joint (Sagl et al., 2019) or when modelling blood vessels (Vorp, 2007).

Sometimes the equations that are defined to control soft tissue behaviour include a viscous term (Huang et al., 2017). Viscoelasticity describes the variation of material response within time containing an elastic and a viscous part. The viscous part can describe creep, when stress remains constant and the deformation increases with time, or relaxation, when the deformation remains constant and stress decreases over time. On the other hand, the elastic response is instantaneous and can be defined using a linear material (Booker & Small, 1977) or a nonlinear hyperelastic material (Kulkarni et al., 2016).

More complex models, including fibres in their formulation, exist for the arterial vessels (Gasser, Ogden & Holzapfel, 2006) or the intervertebral discs (Noailly, Planell & Lacroix, 2011) among others. Despite the complexity of these formulae, which combines the overlay of the stiffness in the preferred directions of the fibres with the hyperelastic formulation of the matrix, the constitutive equation is also nonlinear, and it must be solved following an iterative procedure.

Non-linearities in geometry: buckling

In a linear problem, the equations of equilibrium are formulated in the original undeformed state and are not updated with the deformation. This is common in most engineering problems because the deformations are small enough to avoid differentiating the original geometry and the deformed one. However, there are cases where the deformation cannot be ignored, and we need to include large displacement non-linearities due to the geometrical update during the application of forces: This is the case of buckling.

Buckling implies a sudden change in shape of a body under load because the loss of stability when this load reaches certain critical value (Fig. 4). If a body- such as a column under compression or a plate under shear, for example- is subjected to a gradually increasing load, when the load reaches the critical value, the body may suddenly change shape. Although buckling appears before failure, it can be decisive in the ergonomics of certain biological bodies, limiting the range of forces under which they are able to remain functional. Buckling is caused by nonlinearities in the geometry and can be approached by a linearisation that drives to a bifurcation problem of eigenvalues. Therefore, the linear buckling analysis is done in parallel to a linear elastic analysis. Otherwise, the full nonlinear solution of the point of collapse can be obtained by increasing the load in smaller steps with an iterative method while the geometry is updated to its deformed state. This latter is significantly more computationally expensive but might be more accurate than the linear buckling. In a biomechanical context, buckling can be found when study slender bodies such as the swordfish rostrum (Habegger et al., 2020), the weevil rostrum (Matsumura et al., 2021) or even in bones under compression such as the vertebrae (Williams et al., 2021). Buckling is also considered in humans as a cause of fractures of postcranial bones (Lee et al., 2009).

Figure 4 Buckling.

Deformed shape and displacement of a column under compression loads solved by (A) an elastic linear solution (B) a linear buckling (C) deformed shape and displacement of a squared plate under compression loads solved by a linear buckling and (D) example of buckling in a ruler under compressive forces. F is the compressive load applied at the column and p is the compressive load applied at the plate.

Non-linearities in contacts

Contacts between two bodies are divided between linear contacts and non-linear contacts. Linear contacts can be included in a linear elastic model without modifying the solving mode and keeping the direct solution. It also involves a low computational cost that is simply a function of the size of the finite element mesh (namely, the number of elements and nodes). However, the inclusion of non-linear contacts changes the solving mode to a non-linear solution with an iterative solver, increasing the computational cost of the analysis. Contacts can be described according to the relationship between the two separate surfaces of each body that become mutually tangential in five general different types according to how they can move perpendicularly to each other and how they can move in the tangential plane. In other words, if they are allowed to separate and slide (Fig. 5).

Figure 5 Contacts.

Different types of contact. The labelling of “bonded”, “no-separation”, “rough”, “frictionless”, and “frictional” is according ANSYS 2021. Other FEA packages could use other similar labelling.

Bonded contacts: when separation and sliding is not allowed. It is a linear contact.

No-separation contact: when separation is not allowed but sliding in the tangential plane is allowed. It is a linear contact.

Frictionless contact: when separation and sliding is allowed. It is a non-linear contact.

Rough contact: when separation is allowed but sliding in the tangential plane is not allowed. It is a non-linear contact.

Frictional contact: when separation is allowed but sliding in the tangential plane is controlled by a friction coefficient. It is a non-linear contact.

Frictional contact can be understood as an intermediate status, where sliding in the tangential plane is not free but is allowed and bonded contact is used when we have two bodies that are perfectly joined but they are created or defined as separate bodies during the FEA modelling. For example can be used for defining all the pieces involving a teeth such as the cortical bone, dentine, enamel, pulp and the PDL (Benazzi et al., 2013; Bucchi et al., 2019)

In general, contacts are found in FEA models involving more than one body and the definition of each contact depends on the nature of its behaviour. It can be found in models when studying the carpal bones of the wrist (Gíslason et al., 2017; Püschel et al., 2020a) or the feet (Ito et al., 2022), the ossicles of the auditory system (Marcé-Nogué & Liu, 2020), the intervertebral discs and the vertebrae of the spine (Guan et al., 2019), all the tissues in the hip (Fleps et al., 2018) or the patella (Fitzpatrick & Rullkoetter, 2012), the mandible, the tempomandibular joint and the skull (Sagl et al., 2019) or the interaction between the bodies in the wings of dragonflies (Rajabi et al., 2016b), bees and wasps (Eraghi et al., 2021), among many others. Therefore, contacts can be used to establish relationship between bones or soft tissues. Contacts are also useful when studying occlusal forces during mastication to model the interaction between teeth and food (Skamniotis, Elliott & Charalambides, 2019) or even the impact of eggshells with the floor (Sellés et al., 2019).

Summary: ideas for palaeontologists

FEA is no longer an incipient technique in the fields of palaeontology, anthropology, and evolutionary biology. Instead, it is nowadays a well-established technique within functional virtual morphology toolkit that has been used in more than 750 biological and evolutionary publications between 2005 and 2020 (Tseng, 2021). Most of this works present FEA models without non-linearities. This is not necessarily a problem by itself if the linear approach is sufficient to answer the scientific question of interest. Indeed, many engineering problems can be solved without trespassing the threshold of the linear models. Therefore, this text does not want to spread an incorrect idea regarding the use of supposedly more accurate non-linear models. In fact, the use of linear and not expensive computational approaches without nonlinearities can be certainly useful to understand the behaviour of many biological systems under analysis. For instance, most of the FEA works that include fossils have focused on the study of skeletal elements that can be successfully modelled using linear elastic material properties and solved using a static analysis under small strains and displacements, i.e. without the need of non-linearities. However, although linear models can be used in a broad range of functional works, the aim of this text is to highlight the value of non-linearities when they can be of utility, or they are needed to improve the knowledge we have in fields such as palaeontology and anthropology.

Non-linear soft tissues

Little is known about soft tissue properties in fossils. The direct examination of fossil soft tissues and preserved blood cells is of little value when studying palaeontological remains due to the degradation or the contamination from modern remains (van Dongen et al., 2017). The reconstruction of soft tissues from fossils is an issue that it is unresolved but can be approached through investigating extant relatives to infer the palaeo-physiology of extinct taxa, e.g., via the phylogenetic bracketing approach (Witmer, 1995). Therefore, any FEA models can potentially include an inference regarding soft tissues properties. As an example, cranial sutures are deformable joints between adjacent bones bridged by collagen fibres and there are several works on fossil taxa that have include soft tissues modelling sutures in Tyrannosaurus rex skull (Cost et al., 2020), australopithecines (Dzialo et al., 2014) or in dicynodonts (Jasinoski, Rayfield & Chinsamy, 2009), as well as FEA models of current lizard species (Dutel et al., 2021), Sphenodon (Curtis et al., 2013) or even some mammals (Bright & Gröning, 2011). All of these examples used linear material properties to characterize the elastic behaviour of soft tissues which can be an appropriate simplification if this is validated experimentally (Bright & Gröning, 2011). However a recent diagnosis suggested that the lack of sutures or and inappropriate modelling can result in inaccurate results of stress, strain or deformation (Rayfield, 2019) although it is not clear how the soft tissue can be accurately predicted in fossils (Broyde et al., 2021). It is at this point that the researcher needs, at least, to be aware that a more accurate modelling of these soft tissues should be done using nonlinear material properties, which in turn involve an increase in the computational cost of the model. Unfortunately, soft tissue models in living animals have not been extensively documented, with the exception of biomedical contexts (Tanaka et al., 2000). Hopefully, studies focused on experimentally testing non-linear models in diverse biological taxa will be carried out in the near future.

Plasticity in retrodeformations

Retrodeformation is very common in fossil taxa as the process that produces the original form of the taxon prior to fossil diagenesis when this has been recovered in any deformed way. Deformation in fossils is produced due to a multiple array of taphonomic and tectonic processes. Overburden stress due to the weight of the overlying sediments linearly compacts the fossil from above causing the fossil to break and/or warp. Other causes of fossil deformation include tectonic stresses and sediment cracking. Under the action of these loads, the fossil can break in a brittle manner or can be distorted plastically, preserving the structure of the fossil due to the lack of breakage. Fossils under plastic deformation, where forces applied during time modify the original shape of the bone structure may be restored. Although there are several techniques to virtually restore deformed specimens available without using mechanical equations (Lautenschlager, 2016), it has sense to use methods from mechanics such as FEA that involve forces if one want to infer which was the actual process that drove the fossil to be deformed (Arbour & Currie, 2012; Di Vincenzo et al., 2017). Modelling this phenomenon would require including the nonlinear plastic behaviour of bone, because retrodeformation is a permanent deformation in cortical bone. In this case FEA could be inversely applied by defining the plastic behaviour of the fossil material and then, setting the forces applied at the fossil as the unknowns of the problem. It would allow to answer the question of which forces do we need to apply deformed bodies to recover its original form.

Buckling in slender bones

In palaeontology there are a lot of slender structures that are susceptible to be analysed using buckling. Probably the most common and useful case would be in bones under compression such as the leg bones of large, heavy dinosaurs and mammals. This is because mass is considered as one of the main factors affecting the morphology and osteological adaptation of these bones (Etienne et al., 2020). To understand how these bones are adapted to the heavy weight that they needed to support, evaluation of the maximal stress as a measure of bone strength is not the only informative metric (Hutchinson, 2021). This is because bone may fail without involving fracture. Alternatively, bone could fail through buckling if it is not stiff enough (Currey, 2012). In this case, buckling needs to be considered, because it can cause the collapse of the legs before the fracture of the bone. Usually, buckling reduces the capacity of the strength of the structure because it appears in a lower value than the yield stress and the fracture stress that defines the strength of the material.

If we assume that the leg bones in heavy dinosaurs as slender columns like those from a building, Euler’s critical load is defined as the compressive load at which the column will suddenly buckle (Timoshenko, 1955). This equation can give clues about the relationship between geometrical factors such as the length of the bones or how they are joined to the articulations. Given that the length, material, or boundary conditions cannot be modified from the original model, Euler’s critical force will depend on the second moment of area or moment of inertia. Increasing the value of the critical force implies a modification of the cross-section of the bone through more inertial geometries. Therefore, if we assume the cross-section of leg bones as an annulus, thicker annulus will increase the inertia. But also, if the thickness is kept constant, a broader annulus will increase the inertia of the cross section. This simple consequence can be obtained assuming leg bones with a straight morphology not close enough to the reality, but very useful for the purpose of study. However, in case of analysis of the real and irregular geometry of the bones, the simple formula of Euler cannot be used but the problem of buckling can be solved via computational methods by means of FEA solving an eigenvalue problem. Few works are paying attention to it, discarding the effect of buckling in the morphology of the long bones in living mammals (Brassey et al., 2013). Considering than an eigenvalue problem in a FEA model is not increasing the computational cost of the analysis too much, it would be worth to more exhaustively test if the leg bones of heavy dinosaurs or mastodontic extant mammals are affected by buckling, as was suggested in horses (Currey, 2003). Another case where buckling could be a concern is in bones with a high aspect ratio where the walls are substantially thinner (De Margerie et al., 2005). This could be the case of wing bones (Palmer & Dyke, 2012), and hence buckling should be also explored.

Bone grouping using contacts

Functional implications of fossil bones have been widely studied in fossil taxa using FEA models (Richmond et al., 2005). Depending on the purpose, bones can be studied alone or as a group and the main difference between these two cases is the absence or presence of contacts. When separation between bones is not desired, for example in the analysis of teeth, considering the bonding of the cortical bone, dentine, enamel, pulp and the PDL (Benazzi et al., 2013), the contacts used are linear and it does not require an increase in the computational cost of the solving process. This is something that can be considered when creating FEA models because it allows the inclusion of several bones in the model without nonlinearities.

On the other side, nonlinear contacts allow separation between the bones. Although the inclusion of this contacts implies an iterative solution through convergence, it may be necessary to implement when a group of bones need to be studied together such as the carpal bones of the wrist (Gíslason et al., 2017) or the bones of the foot (Ito et al., 2022). This has been done extensively in biomechanical models of living primates; therefore, it should be considered in other FEA works in the field of the palaeontology and anthropology. In fact, the literature is full of biomechanical analysis of kinematics and dynamics of solid bodies where bones of fossils are grouped to study its performance (Sellers et al., 2017; Bishop et al., 2021). Therefore, it makes sense when creating FEA models, to include more than one bone in the model if it can be useful for the desired analysis despite increasing the computational cost of the solution. Also when the contact between bones is through articular cartilage, the contact can be defined between cartilages that are also in contact with the bone (Püschel et al., 2020a).

Models with shells, plates, beams, springs, and trusses

Finally, although this is not related with the use of a nonlinear iterative solving, the use of other kind of elements other than solid elements can have a great advantage when dealing with nonlinear models. This is because they provide a useful way to reduce the number of elements and nodes of the FEA mesh and, consequently, a reduction of the time spent solving the equation in each iteration. Hence, a nonlinear model will particularly benefit from the use these elements.

The use of shell elements to model cortical bone in morphologies that can be assumed as thin and with a constant thickness, such as carpal bones or talar morphologies, require a lower number of elements and nodes because there is only one mesh layer. Using solid elements in the same morphology would need at least four or five layers of elements would along the thickness to properly build an adequate mesh to accurately capture the results. This was used in an analysis of carpal bones (Gíslason et al., 2017) to model both the cortical bone and the articular cartilage, reducing significantly the number of elements to allow a smooth non-linear solution, due to the presence of non-linear contacts. The same example uses non-linear spring elements to model the behaviour of the ligaments. This decision is also in favour of not increasing the number of nodes and elements of the model, because spring or truss elements can be defined using only one element with the origin and final nodes. In this manner, the model avoids the inclusion of a three-dimensional geometry modelled with solid elements for each ligament, which would exponentially increase the number of nodes and elements in the mesh and consequently, increase the computational cost of the solution.

When creating FEA models of fossils and considering the inclusion of some of the non-linearities it is a good option to evaluate if the use of simpler elements can reduce computational cost. Although researchers should be aware of the potential ramifications of simplifying their models, it is also true that any model will necessarily not represent a literal representation of reality. Instead, the requirements necessary to answer the research question of interest should always be kept in mind when making decisions about model complexity.

Summarizing, this text highlighted the usefulness of non-linearities in FEA palaeontological and anthropological models in spite of increasing their complexity and the computational costs. Nowadays, most of the commercial and non-commercial FEA packages include the resolution of non-linear problems in their capabilities, and they also documented with tutorials and examples how to deal with them. Therefore, the main aim of this review is to provide a road map for the next generation of palaeontologists, anthropologists, and functional morphologists by showing them unexplored ways that could a profound impact in finite element analysis and how they can explore these methods further.

The author wants to thank Lluis Gil and Josep Fortuny for readings and commenting earlier versions of the manuscript, as well as acknowledging Thomas A. Püschel for reviewing and proof-reading the text. The text was also substantially improved thanks to the review done by Emily Rayfield and another anonymous reviewer.

Additional Information and Declarations

Competing Interests

Author Contributions

Data Availability

The author declares that they have no competing interests.

Jordi Marcé-Nogué conceived and designed the experiments, performed the experiments, analyzed the data, prepared figures and/or tables, authored or reviewed drafts of the article, conceived the text, and approved the final draft.

The following information was supplied regarding data availability:

This is a literature review; there is no raw data or code.

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
