# Peer review of "One step further in biomechanical models in palaeontology: a nonlinear finite element analysis review"

_PeerJ, doi:10.7717/peerj.13890_

## Round 0.1 · original submission · Major Revisions

Two reviewers have given extensive, constructive, insightful comments on the manuscript, and are supportive of it; I agree. The main concern is that validation needs to be addressed more openly, and especially in light of the uncertainty of whether non-linear approaches add precision and accuracy. It is important to satisfy both reviewers in your revision and have those revisions be in the main manuscript, not solely in the Rebuttal to reviewers. But this will clearly be a useful review so thank you for submitting it.

Reviewer 1 ·

Basic reporting

The basic reporting is mostly strong, however, there are quite a few grammatical problems that do need to be addressed to make the article clearer. I recommend that the article is proof-read before publication.

Experimental design

No comment

Validity of the findings

No comment

Additional comments

Summary:
This paper presents a review of ways in which non-linear approaches can (and arguably should) be incorporated into zoological, and by extension paleontological, FEA. I think that such a review is timely and will be very useful to other workers. Non-linear FEA is hardly ever used in zoo/paleo modelling, and while it is often unnecessary, sometimes its inclusion will be essential. This review provides an overview of these situations.

While I like this work and think that it could be an important reference for many people, at the moment, I think it is something of a missed opportunity. This is because I disagree that the main reason that biologists don’t use non-linear FEA is because of the computational cost, although this can certainly be a factor. Instead, I think the main reason is a widespread ignorance about how FE problems are set up and solved, and that people are generally unaware that they could or should be using non-linear methods. Therefore, I think that this review could do more to educate readers on these fundamentals. If this was added, I believe that the value of the paper would increase substantially.

More detailed comments:
I find that the paper is well-written overall, however, there are quite a lot of small grammatical mistakes (too many for me to list here) that can obscure your meaning. It’s mostly things like missing articles, subject/verb disagreements, or synonyms with subtly different meanings to the one that is intended; they’re small things, but given that this is quite a technical topic where logic is paramount, they should be fixed. Therefore, I suggest that the manuscript would benefit from further proof-reading, either by a professional or a colleague with a pedantic love of grammar.

A few times: The word ‘stablish’ appears, and it is unclear whether this is supposed to say ‘establish’ (verb: to set up), or ‘stable-ish’ (not a real word, but would be commonly understood to mean a configuration of bodies that is somewhat stable, but wobbly). I expect it’s the former, because on line 241 you say “to stablish”, but until then I’d assumed it was the latter. Given that this word appears in the context of setting up contact problems it needs to be clarified, as either interpretation could work.

Introduction, first paragraph: While I don’t find that there’s anything particularly wrong with what is written here, I think you need to expand your text by stepping back even further and explaining more plainly how FE models are constructed, and where the linear/non-linear equations are brought into this process. As it stands, you seem to be assuming quite a lot of prior knowledge of FEA, and that biologists will have an intuitive grasp of linear vs. non-linear problems: in my experience this is not often true. My overwhelming suspicion is that most paleo FE users make linear models not because they are trying to keep the model simple, as you suppose, but rather, they are genuinely unaware of the differences between linear and non-linear FEA and why one might be more appropriate than the other for their analysis. It just so happens to be the case that most problems that have been looked at by paleontologists to date CAN be treated as static, linear problems, so few have had any reason to explore the methods further. You can therefore use this paper as an opportunity to send out a warning as well as highlighting the wider range of questions that can be asked with FEA.

You have a good explanation of what a linear model is on line 190, so perhaps you could move this forward to your introduction? Defining both linear and non-linear models in plain language right at the start would be helpful for your readers. You could perhaps also include some brief examples (without leaning heavily on physics or engineering terminology) of what a non-linear model for zoo/paleo FEA might look like, and explain how the key variable that’s different between linear/non-linear models is how certain aspects of the input/output change with time. I know that you address this in detail later, but I think you’re more likely to retain readers if they can instantly see how this issue is applicable to their studies.

L70: I’ve never thought of Anderson and Westneat’s (2007) model as a finite element model of trusses! I suppose it is the equivalent of that, in that there are four trusses joined at nodes, but the authors never called their model FEA, and the purpose of this model (per my reading) is to understand the rotational motion of those bars based on rigid body displacement. Given that the nodal displacements in this model are large, and that large displacement problems are a specific candidate for using non-linear models, I wonder if citing this paper as an example of 1D FEA isn’t going to get too confusing for people? Because it isn’t something that people are going to envision as FEA, even though technically it is a model made from four 1D elements connected by nodes. That said, I don’t have a better suggestion; I’m not aware of other models that have used trusses in biomechanical analyses.

L88-90: “These kinds of elements … cannot be called as 2D elements because they use the equations of plane elasticity”. This needs further expansion for it to make sense; if these elements are not 3D and not 2D, then what are they? It follows that they are not 3D if they are defined and solving only in the x-y plane, but as this is written it doesn’t explain why one can’t call plane elements 2D elements. Do you mean that they are not 2D in the sense that they are defined with a (small) third dimension thickness? Don’t most commercial solvers still list these under 2D elements?

Fig 2: You need to use the caption to define and explain the letters and colors in this figure. If force is F and displacement is u, then I’m assuming K is some sort of stiffness modulus? I’m also assuming that the orange line is an approximation of the response, the true version of which is the black line (and therefore i represents the iterative steps needed to converge on it?). Without defining all of these parts, the figure is more confusing than it is helpful. (The same comment applies to figure three, but with more letters to define. While many people are likely to be familiar with an elastic-plastic stress/strain curve from high school, the hyperelastic one will be less familiar and needs the terms defining – is W the strain energy function?)

L123: If this review is pitched at biologists, you might have to help them through concepts like Hooke’s Law, which they might be unfamiliar with, or have not encountered for a long time. Here, instead of “The relationship between stress and strain is not following the linear Hooke’s law”, you could say “When a material is non-linear, the strain it experiences is not proportional to the stress applied”. You could follow this with an “i.e. the material does not conform to Hooke’s Law” if you wanted to. I think the point I’m trying to make is that it’ll be easier for non-physicists/engineers if you lead with a simple explanation.

L127: Similarly, you should probably also explain what the small strain theory is. If people are unaware of this assumption, they’re going to be unaware that they’re violating it in certain models, and I think this is behind a lot of the inappropriate use of linear models in bio FEA.
L165: Can you give a more straightforward description of hyperelasticity first, before defining it precisely? For example, something like “A hyperelastic material is one that shows extreme elastic behaviour, in that it is able to return to its original shape even after experiencing very high strains of up to X%”

L166: Can you explain what a strain energy function is? My understanding is that it refers to the elastic potential energy built up in the structure, but beyond that I’ll admit not really knowing how it works.

L167: “…it uses the large deformation theory described above”. Except you didn’t really describe the large deformation theory, beyond “the deformed and undeformed configurations of the body under analysis are notably different”. This doesn’t really explain what the theory is, and all that was done in the section above (L125-131) was state that it should be used when the assumptions of the small strain theory are violated, which as per my last comment, also was not described.

L199: I find that many non-engineers are unfamiliar with the concept of buckling, and struggle to understand it until they’re given an example that they will have encountered in day-to-day life (at which point they recognise it instantly). I don’t think that figure 4 really helps to elucidate things either, because if people don’t appreciate the difference between bending and buckling (and most biologists won’t), then Fig. 4b will just look like bending. I know that your point with the figure is to illustrate the differences between the linear and non-linear solution rather than how buckling is different to bending, but I fear that this could get lost if people are confused about what they’re looking at. Could you show a photograph of a buckled structure in Fig. 4, alongside a common example in the text? My go-to for a buckling column is how a drinking straw or soda can will suddenly fold if you compress it lengthways. Is there an everyday example that you could give for a sheet?

L235-241: This is a very long sentence that doesn’t resolve. Please revise.

L267: “…which can be an appropriate simplification if this is validated experimentally.” I think this is an important observation, and I am pleased that you have made it. I wonder if it’s worth pointing out that none of the validation studies that came out in the period where they were popular (c. 2005-15) considered whether non-linear models were more accurate than linear ones (at least, not to my knowledge). Of course, many of these models were deemed to be ‘good enough’ and wouldn’t be candidates for non-linear properties anyway, which is the point that you’re making here; however, would a call for some more experimental testing of non-linear models in non-medical, biological FEA be prudent? I feel like you’re getting at this in your closing sentence (L280-282), but you don’t quite say it. Indeed, is it sensible for paleontologists to use the methods you describe before any non-linear validation studies have been performed?

L285: “fossildiagenesi” Should be diagenesis?

L303: Another obvious one that springs to mind is the wing bones of pterosaurs; although their mass is likely to be lower than the other examples that you have given, the bone walls are substantially thinner, and their high aspect ratio makes buckling a real concern. Colin Palmer has done some work on this that you could look up, although I can’t recall the exact references right now (and I don’t remember FEA being used).

·

Basic reporting

The review considers the use of non-linear approaches in finite element modelling, with the aim of introducing the potential of these approaches to a palaeontological audience. This is clear in the introduction, as is the motivation of the study. Later, the 'palaeontological audience' is more broadly interpreted as palaeo, archaeology, and zoology, which is, in reality, the audience the review is aimed at.
There is scope for a review of this nature. As the author points out, most studies and most previous reviews have focused on linear static methods in the bio- and historical sciences. Yet in medical sciences, non-linear approaches are more commonplace. The review will be of interest to the field.
For the most part a description of the mathematical and computatinal underpinnings are matched to the relevant biological structure or approach (although see below where these links could be stronger or clearer).

Experimental design

In a short review such as this it's impossible to capture the breadth of the field. However the examples chosen do span taxonomic groups and different approaches. I don't have any concerns about this. The references seem adequately cited. Organisation seems logical.

Validity of the findings

So this is where I have the majority of comments. My overwhelming feeling when finishing reading the manuscript was, we can do all of this, but *should* we do all of this, and do we *need* to do all of this? This is because one thing that's missing from the whole review is a discussion of validation. The non-linear approaches are introduced, there’s a discussion of which materials, tissues or structures and loading scenarios might lend themselves to a non-linear approach – I have no qualms about this. But, do we need to introduce this level of complexity to get the results we need? There are some mentions of this in the text, but the danger is that a reader may interpret this paper to think that one must apply these approaches to get a more accurate and realistic result. But is this really the case? In most instances we don’t know. Take the PDL as an example. We could define the PDL as a contact surface, but do we need to define this as a contact surface to get the most accurate results? Do we need to actually introduce such complexity into our models? Likewise, we could introduce muscles as non-linear features, but actually some elastic linear studies have got really quite good results using subject specific models, detailed material properties and applying muscle forces directly to the model. Take Panagiotopoulou et al. 2017 (https://www.sciencedirect.com/science/article/pii/S0944200616301805) as a good example of a linear elastic model where computational strain values that match those measured in vivo with a high degree of accuracy. Explores the role of the PDL again using a linear elastic noncontact approach (Panagiotopoulou et al. 2019) https://www.frontiersin.org/articles/10.3389/fbioe.2019.00269/full
This review needs to acknowledge the lack of understanding of how much introducing non-linear methods improves accuracy or predictability of models.

Sections on plastic deformation (line 151 onwards) and buckling (line 189 onwards). These sections need to acknowledge that plastic behaviours in materials such as dentine or enamel represent the extremes of the loading environment rather than everyday behaviour. Likewise simulations of buckling in structures such as limb bones would also represent extreme loading conditions at the point of critical failure. Neither would be useful simulations to understand everyday loading behaviours. Further on the point of buckling, I'm not convinced that a rigid upright column is the appropriate model to represent a multi-jointed musculoskeletal system such as the limb. Why for example might it be useful to test if the bones of dinosaurs or mastodons are affected by buckling? Surely this would tell us the maximum loads that could be experienced but equally this could be achieved simply by running simulations of individual limb bones and seeing which one reaches critical failure first? In both this context of plastic deformation and buckling the review needs to be careful not to mislead the reader that such non-linear simulations would represent everyday behavioural functions.

More specific comments:
Line 19 - assuming linear elasticity in bone is not a simplification for most loading scenarios (avoiding large loads).
Lines 23-25 – also we use linear models because we understand the properties of these materials better than those with non-linear properties? And, we’ve performed validation studies in some models that show that the results are appropriate for the question of interest.
Lines 76 to 104. There’s some repetition in these three paragraphs (see comments on my PDF).
Lines 279 to 282. Picking up on the question of muscles again, if it's not clear how soft tissues can accurately be predicted in fossils, then surely we should be avoiding modelling these soft tissues using non-linear material properties because this has the potential to introduce more error to our models?
Line 330 – section 5.4. I think the section on using contacts could be more nuanced. For example, I think it's a good idea to include contact surfaces when modelling multi bone structures with joints and range of motion, such as the carpal bones of the wrist or the bones of the foot (as examples presented here). However, going back to my original point about the PDL, do we need to model adjacent materials that lack articulations as contact surfaces? Studies that have tried to model sutures in different ways for example using contact surfaces or springs, have shown variable results. It goes back to the point about whether introducing more complexity is really worth the extra time and effort and computational power and if this is going to produce a more accurate result, and indeed whether a more accurate result is actually warranted by the question at hand. A good example of this is modelling fossils, where we do not know material property values or the true extent of soft tissues and therefore muscle input forces with a high degree of accuracy. Is it therefore necessary to introduce other complex features when we know our results will not represent the absolute stresses and strains student structure in life?
I have a few different comments on the figures. I’ve added these to pdf.

Additional comments

N/A

---

## Round 0.2 · Minor Revisions

One reviewer has provided very helpful recommendations for the main text. I suggest that the title be changed to omit the "One step further in palaeontology" as every paper in palaeontology is one step further in some way, and that part of the title is too vague/general to tell readers anything about the paper's content. I agree on buckling - its relevance to larger animals is less certain. As usual, Alexander (and Currey) had something to say about this-- "The thickness of the walls of tubular bones", J. Zoo!., Lond. (A) (1985) 206, 453-468. Please revise the paper according to all of the recommendations- thank you!

·

Basic reporting

This is a useful review and one that I would like to see published. Following the first review, my main concern was that the author make it clear that they are showcasing the overlooked potential and importance of non-linear FE models rather than appearing to advocate without bias for their use. In this revised version, I am happy to see that the author has modified the text to make the review closer to the former rather than the latter. This is particularly clear in the Abstract and at the end of the Introduction.

As it stands, I am content that this revised manuscript serves the purpose of outlining the usefulness of non-linear models. There are still a few queries and adjustments I would like the author to make before publication - outlined below.

Experimental design

No comment.

Validity of the findings

Line 32 - when you say "mathematical nonlinearities are natural in physical models and the assumption of nonlinearilty...[etc]" by 'physical models', do you mean organisms and biological systems? When I see the word 'models' I think of mathematical/computational constructs, but I think you mean something different here?

Line 38 onwwards - if palaeontologists are unaware of non-linear models, then they cannot ignore something they are not aware of. I suggest instead of ignore, phrase this as something like: "...most palaeontologists do not have a deep understanding of how FEA problems are defined and solved, and many are not aware of the potential of non-linear modelling."

Line 41 onwards: "Including non-linearities in palaeoanthropological models can improve the results obtained in skulls when modelling skulls by considering sutures, thus making the behaviour of the model closer to reality." If you want to convince palaeo/anthro people to use non-linear models, you need to add a reference to show how this is indeed the case (maybe from the human/medical literature?).

Line 103 onwards: "In most of the biological models modelling bone structures, shells have been the prefered option." Preferred on what basis? Accurate geometry or performance? I can see how this will work for a thin later of cortex and underlying cancellous bone, but surely shells are not preferred when dealing with thick cortical bone for example? This sentence needs clarifying why shells are preferred and whether this is the case for all bony structures.

4.2 Non-linearities in Geometry: Buckling - this section should refer to Figure 4 I presume (not figure 3).

4.3 Non-linearities in contacts - likewise in this section should refer to Figure 5.

5.1 Non-linear soft tissues - it would be great at the end of this section to include some references of biomedical models that have succesfully included non-linear materials - to guide the reader to further recommended reading.

Regarding buckling - the modified figure is helpful. And I understand how buckling may be important for the slender long bones of some birds and pterosaurs. Arthropod cuticle could be another good model strucutre. However I'm still not convinced that buckling is an important selective criterion for the thick long bones of high body mass dinosaurs and proboscideans. Are buckling injuries recorded in the human and veterinary literature? I need some more evidence before dinosaurs and mammals are used as exemplars.

Additional comments

Line 80: kind should be kinds

Finally - the review ends abruptly. It would benefit from a short conclusion paragraph. I suggest the author add this to summarise their ideas and suggestions. (I should have suggested this in the first review - it was an oversight on my part).

---

## Round 0.3 · accepted · Accept

I have checked these revisions and they are sufficient. I am not so sure about buckling being a general constraint on bone design beyond thin-walled bones. The logic doesn't seem too stron, e.g. if thin-walled then bucking is important, if thick-walled then they evolved that way to avoid buckling (hence it is still important), so it sort of becomes unfalsifiable/circular. But this is something future literature could explore.